# Optimization of Ohmic Contact to Ultrathin-Barrier AlGaN/GaN Heterostructure via an 'Ohmic-Before-Passivation' Process

**Yuan Ji** [1,2] , **Sen Huang** [1,2,*] , **Qimeng Jiang** [1,*] , **Ruizhe Zhang** [1,2] , **Jie Fan** [1] , **Haibo Yin** [1,2] , **Yingkui Zheng** [1,2] , **Xinhua Wang** [1,2] , **Ke Wei** [1,2] and **Xinyu Liu** [1,2]

[1] Institute of Microelectronics of the Chinese Academy of Sciences, Beijing 100029, China; jiyuan@ime.ac.cn (Y.J.); fanjie@ime.ac.cn (J.F.); yinhaibo@ime.ac.cn (H.Y.); zhengyingkui@ime.ac.cn (Y.Z.); weike@ime.ac.cn (K.W.); xyliu@ime.ac.cn (X.L.)

[2] University of Chinese Academy of Sciences, Beijing 101408, China

\* Correspondence: huangsen@ime.ac.cn (S.H.); jiangqimeng@ime.ac.cn (Q.J.)

**Abstract:** Non-recessed ohmic contact resistance ($R_c$) on ultrathin-barrier (UTB) AlGaN(<6 nm)/GaN heterostructure was effectively reduced to a low value of 0.16 $\Omega \cdot$mm. The method called the 'ohmic-before-passivation' process was adopted to eliminate the effects of fluorine plasma etching, in which an alloyed Ti/Al/Ni/Au ohmic metal stack was formed prior to passivation. The recovery of 2-D Electron Gas (2DEG) adjacent to the ohmic contact was enhanced by composite double-layer dielectric with AlN/SiN$_x$ passivation. It is found that the separation between the recovered 2DEG and the ohmic contacting edge can be remarkably reduced, contributing to a reduced transfer length ($L_T$) and low $R_c$, as compared to that of ohmic contact to the AlGaN(~20 nm)/GaN heterostructure with a pre-ohmic recess process. Thermionic field emission is verified to be the dominant ohmic contact mechanism by temperature-dependent current-voltage measurements. The low on-resistance of 3.9 $\Omega \cdot$mm and the maximum current density of 750 mA/mm with $V_g$ = 3 V were achieved on the devices with the optimized ohmic contact. The non-recessed ohmic contact with the 'ohmic-before-passivation' process is a promising strategy to optimize the performance of low-voltage GaN-based power devices.

**Keywords:** AlGaN/GaN heterostructure; non-recessed; ultrathin barrier (UTB); ohmic contact; transfer length



## 1. Introduction

The GaN-based high electron mobility transistor (HEMT) has demonstrated excellent performance in power conversion and radio frequency (RF) power amplification, owing to its high breakdown electric field, electron-saturation velocity, and especially high density of polarization-induced 2-D electron gas (2DEG) [1–3]. Researchers have been sparing no effort to reduce the ohmic contact resistance ($R_c$) of the HEMT, as it is essential to achieve a high-saturation current, low on-resistance for power devices, and high frequency as well as high efficiency for RF devices [4–7]. Among the many arts to reduce $R_c$, such as n⁺-GaN regrowth [8–11], Si ion implantation [12–14], pre-ohmic recess are commonly adopted to form lower $R_c$ for AlGaN/GaN HEMTs with conventional barrier thickness (named as CB-AlGaN/GaN HEMTs in the following discussion) as the tunneling distance between the ohmic metal and 2DEG channel can be effectively reduced [15–17]. However, recess etching of the AlGaN barrier, from a starting thickness of 15~30 nm down to several nanometers, is challenging for most etching equipment. Schemes of non-recessed ohmic contacts, as a result, are highly attractive for CB-AlGaN/GaN HEMTs.

The ultrathin-barrier (UTB) AlGaN/GaN heterostructure, by virtue of its naturally depleted 2DEG channel, has been developed for the fabrication of normally-OFF, high-threshold voltage uniformity AlGaN/GaN HEMTs [4,18]. It is also a promising technology

platform for non-recessed ohmic contact. Even though 'ohmic-before-passivation' with $SiN_x$ passivation using PECVD is usually a conventional process for CB-AlGaN/GaN heterostructure. However, in previous experiments, the 'passivation-before-ohmic' process with LPCVD-SIN$_x$ passivation is often adopted to recover 2DEG for UTB-AlGaN/GaN HEMTs before the development of charge-polarized AlN passivation [4,16,19–21]. In such a 'passivation-before-ohmic' process, the LPCVD-SiN$_x$ passivation in the ohmic region should be removed by fluorine-based plasmas first, which would result in negatively charged F$^-$ ions in the contacting edge [17]. For this reason, the R$_c$ was difficult to reduce to below 0.5 $\Omega \cdot$mm in previous experiments [4,16,19–22].

In this work, with the development of charge-polarized AlN passivation for 2DEG recovery, the 'ohmic-before-passivation' process was attempted to get rid of the influence of F$^-$ ions on ohmic contact with UTB-AlGaN/GaN heterostructure associated with the 'passivation-before-ohmic' process. AlN/SiN$_x$ stacked passivation is grown after high-temperature ohmic annealing to avoid metal overflowing at the edge of the dielectric stack. A significant reduction of R$_c$ to 0.16 $\Omega \cdot$mm is realized. Temperature-dependent current voltage and microstructural characterizations revealed a remarkably reduced transfer length (L$_T$) as well as metal-to-2DEG tunneling distance owing to different metal profiles, as compared with the recessed ohmic contact to CB-AlGaN/GaN heterostructure. The low on-resistance of 3.9 $\Omega \cdot$mm and the maximum current density of 750 mA/mm with V$_g$ = 3 V were achieved on the devices with the optimized ohmic contact. The non-recessed ohmic contact with the 'ohmic-before-passivation' process is a promising method to optimize the performance of UTB-AlGaN/GaN heterostructure for low-voltage GaN-based power devices.

## 2. Device Structure and Fabrication

UTB- and CB-AlGaN/GaN heterostructures used in this work were grown by metal-organic chemical vapor deposition (MOCVD) on Si substrates. They consist of a thick GaN buffer layer, a GaN channel layer of around 50 nm, and a UTB or CB AlN/Al$_{0.25}$Ga$_{0.75}$N/ GaN (cap) barrier layer. The prescribed thicknesses of the GaN cap, Al$_{0.25}$Ga$_{0.75}$N layer, and AlN interface enhancement layer for UTB and CB are about 1, 4, 1 nm, and 1, 20, 1 nm, respectively. Low-power Cl$_2$/BCl$_3$ hybrid plasmas were used for the pre-ohmic recess of the CB-AlGaN/GaN heterostructure before ohmic mentalization. Afterward, both the recessed CB-AlGaN/GaN and un-recessed UTB-AlGaN/GaN heterostructures were then wet-treated in a diluted HCl, and a Ti/Al/Ni/Au metal stack was evaporated to serve as the ohmic contacts. The ohmic metal system has four layers. The first layer, Ti, is a refractory metal; the main purpose is to react with AlGaN to generate low-resistance nitride such as TiN. The second layer is a low melting point of Al; the purpose is to accelerate the reaction of Ti with AlGaN. The third layer of Ni needs to be a good barrier layer, which can prevent the first and second layers of metal and the fourth layer of metal diffusion of each other. Au, as the fourth layer of metal, mainly plays a role in preventing oxidation. After lift-off, both samples were subjected to an annealing process at 810 °C for 50 s in a N$_2$ atmosphere. Although the melting point of Al and Au is low, the downward diffusion of Au can be prevented due to the presence of a barrier layer, and the low melting point of Al can accelerate the formation of low-resistance TiN. However, the low melting point is possible to improve the surface roughness. This annealing condition is an empirical condition that has been experimentally explored, which comprehensively considers the effects of annealing on the degree of reaction and surface roughness of low-resistance nitrides. A passivation stack of the AlN/SiN$_x$ (4/40 nm) layer, with the first 4-nm AlN layer grown by plasma-enhanced atomic layer deposition (PEALD) and the 40-nm SiN$_x$ layer by PECVD, was grown to recover 2DEG in the UTB sample. Schematic diagrams of the structure of both ohmic contacts made using the above process are shown in Figure 1. The 2DEG density in the PEALD-AlN/PECVD-SiN$_x$ passivated UTB sample, as determined by mercury-probe capacitance-voltage measurements, was $8.55 \times 10^{12}$ cm$^{-2}$, which was lower than a value of $1.10 \times 10^{13}$ cm$^{-2}$ in the CB sample without the passivation. High-energy

Ar$^+$ ion implantation was then used to isolate the device. Finally, Ni/Au was evaporated to serve as a gate.

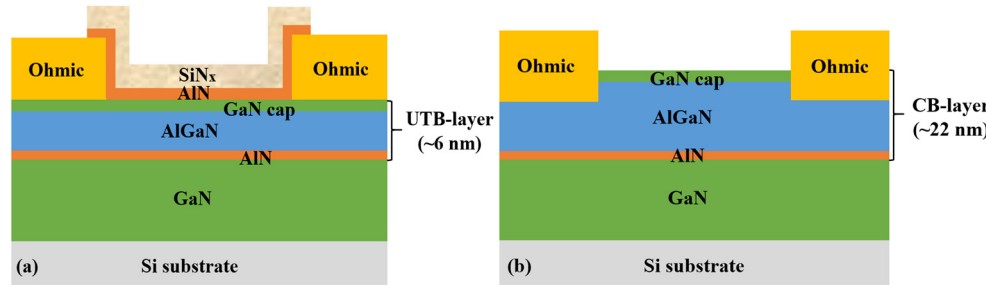

**Figure 1.** Schematic diagrams of the structure of the ohmic contact on (**a**) UTB−AlGaN/GaN heterostructure and (**b**) CB−AlGaN/GaN heterostructure.

## 3. Results and Discussion

The R$_c$ and sheet resistance (R$_{sh}$) for the PEALD-AlN/PECVD-SiN$_x$ passivated UTB- and unpassivated CB-AlGaN/GaN heterostructure were determined using the circular transmission line method (CTLM). The four-probe approach was adopted to eliminate the parasitic resistance of probes. It is interesting to note that ohmic contact only formed after the passivation for the UTB-AlGaN/GaN heterostructure. It is probably due to the recovery of 2DEG at the ohmic contacting edge so that the tunneling probability between ohmic metal and the 2DEG channel can be significantly enhanced. For ohmic contacts, the effective electron conduction is concentrated at the source-drain edge due to the current crowding effect, which has also been reported in previous studies [17]. Figure 2a,b show the current-voltage (I-V) characteristics of ohmic contacts to the two heterostructures. Good linear behavior is observed for all designed spacings, ranging from 8 to 36 μm. The evolution of measured resistance with the contacting spacings is shown in Figure 2c,d for both samples. Note that the spacing variations caused by their process difference are all considered in extractions of R$_c$ and R$_{sh}$.

Upon linear fitting, the R$_c$ and R$_{sh}$ for the UTB samples are determined to be 0.16 Ω·mm and 299.57 Ω/sq; while for the CB heterostructure, they are 0.53 Ω·mm and 276.44 Ω/sq, respectively. The lower R$_{sh}$ of the CB sample is owing to its higher density of 2DEG outside the ohmic contact area than that of the UTB ones. Nevertheless, the R$_c$ of the UTB sample is remarkably lower than that of the CB ones. The ohmic contact mechanism is further analyzed by plotting the change of specific contact resistivity ($\varrho_c$) with temperature, as shown in Figure 3a. For both UTB and CB heterostructures, the extracted $\varrho_c$ decreases with temperature, which can be well-simulated by thermionic field emission (TFE) [23]. It is consistent with the ohmic contact mechanism reported on the AlGaN/GaN heterostructure [24], n-type GaN [25], and p-type GaN [26]. Two crucial parameters, $\Phi_B$ and N$_D$, can be acquired with the TFE model fitting shown below [17],

$$\varrho_c = \frac{1}{qA^*} \frac{k^2_B}{\sqrt{\pi(\Phi_B + E_n)E_{00}}} \cosh\left(\frac{E_{00}}{k_BT}\right) \cdot \sqrt{\coth(\frac{E_{00}}{k_BT})} \times \exp\left(\frac{\Phi_B + E_n}{E_0} - \frac{E_n}{k_BT}\right), \quad (1)$$

where

$$E_0 = E_{00} \cdot \coth(\frac{E_{00}}{k_BT}), \quad (2)$$

and

$$E_{00} = \frac{qh}{4\pi} \sqrt{\frac{N_D}{m^*\varepsilon}}, \quad (3)$$

where A$^*$ = 4πm$^*$k$^2_B$/h$^3$ is the effective Richardson constant, m$^*$ is the effective mass of 2DEG, $\varepsilon$ is the dielectric constant of AlGaN, N$_D$ is the electron carrier concentration, and $\Phi_B$ is the height of the Schottky barrier between AlGaN and the ohmic metals, and

$E_n$ is the energy difference between the conduction-band edge and Fermi level at the AlGaN/GaN interface.

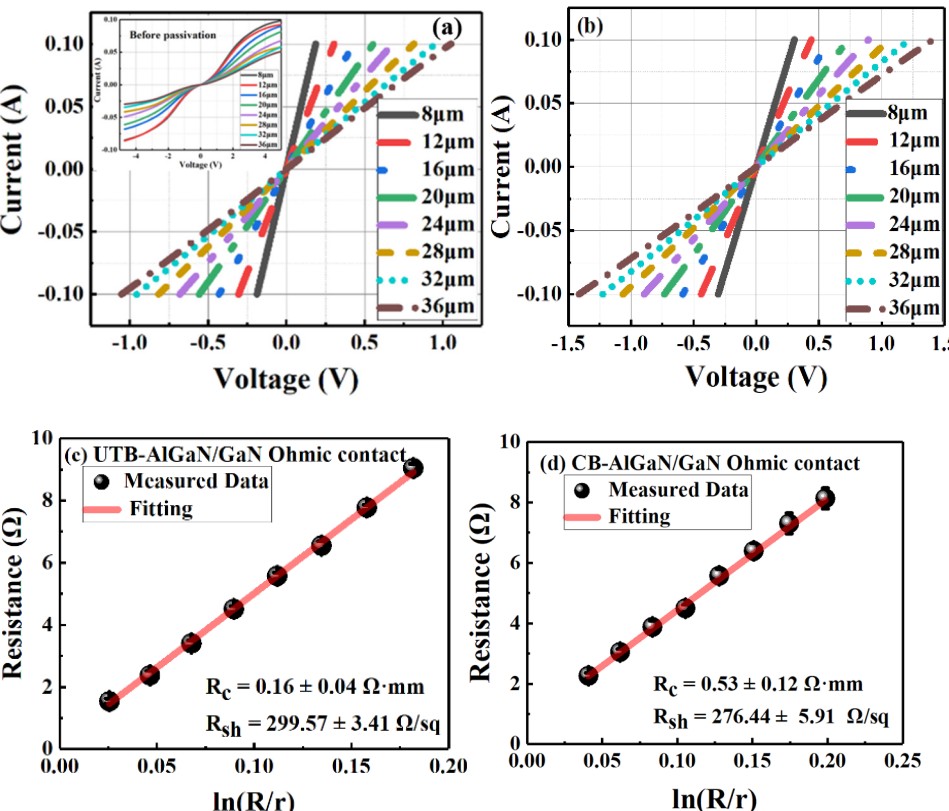

**Figure 2.** I–V characteristics of ohmic contacts to (**a**) UTB−AlGaN/GaN heterostructure and (**b**) CB−AlGaN/GaN heterostructure. Evolution of ohmic resistances with ohmic-contact spacing in samples with (**c**) UTB−AlGaN/GaN heterostructure and (**d**) CB−AlGaN/GaN heterostructure.

The TFE fitting produces a $\Phi_B$ of 0.45 eV and an $N_D$ of $1.59 \times 10^{19}$ cm$^{-3}$ for the CB heterostructure, as opposed to values of 0.61 eV and $1.10 \times 10^{20}$ cm$^{-3}$ obtained in the UTB ones. Similar orders of magnitude of $N_D$ (~$10^{19}$ cm$^{-3}$) and $\Phi_B$ (0.5 eV) have also been reported in ohmic contacts to CB-AlGaN/GaN heterostructure [23]. The lower $\Phi_B$ in the CB sample, as compared with that of the un-recessed UTB ones, may result from the recess-enhanced interface reaction between Ti and AlGaN to promote the formation of low-resistance nitride TiN. On the contrary, thanks to the recess-free characteristic of the UTB-AlGaN/GaN heterostructure, a relatively intact AlGaN barrier was reserved, contributing to a higher $N_D$ than that of the CB sample. Usually, the current does not flow through the entire ohmic metal, and most of the current is collected at the edge of the ohmic metal because the potential at the edge of the electrode is the highest, which is called the current edge effect. As the distance increases, the potential decreases exponentially. Therefore, $L_T$ is defined as the length through which current flows when the current or voltage drops to $1/e$ of the edge of the contact area. $L_T$ reflects the transmission path of current and is also very important for subsequent studies. Its expression can be expressed as below:

$$L_T = \sqrt{\frac{\varrho_c}{R_{ch}}}, \tag{4}$$

The transfer length ($L_T$) of ohmic contacts was also plotted in Figure 3b with the temperature. A remarkable reduction of $L_T$ is observed in the UTB heterostructure as compared to that of the CB ones, which suggests a more concentrated current path at the ohmic-contacting edge. The mechanism for the faster drop of $L_T$ at below 75 °C, as for the UTB sample, need further investigation.

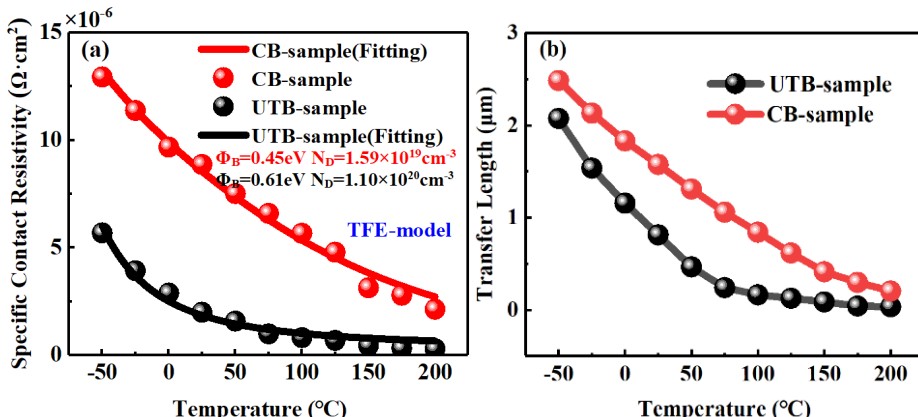

**Figure 3.** Measured (symbols) and modeled (lines) temperature−dependent (**a**) specific contact resistivity and (**b**) transfer length of ohmic contacts to UTB−AlGaN/GaN heterostructure and CB−AlGaN/GaN heterostructure, respectively.

Transmission electron microscopy (TEM) and energy dispersive spectroscopy (EDS) mapping were adopted to analyze the profile and element diffusion between ohmic metal and the AlGaN/GaN heterostructure for the two types of samples (Figure 4). For the UTB heterostructure, there is a slight and uniform reaction between the metal stack and the AlGaN barrier (Figure 4a) [26], in contrast to the more severe interface reaction in the CB ones shown in Figure 4b. In the ohmic contact region, the barrier thickness of the UTB structure is 6 nm, and that of the CB is 3.6 nm. In our previous experiments, we also made ohmic contact on the same UTB-AlGaN/GaN heterostructure without etching with the "passivation-before-ohmic"process, the $R_c$ of which is 1.57 $\Omega \cdot$mm [17]. Therefore, compared to the slight difference in barrier layer thickness, the influence of the "ohmic-before-passivation" process on ohmic contact is more obvious. Owing to their process difference, different metal/2DEG angles are clearly observed in the UTB and CB samples. The CB sample features a quite gentle slope angle of 156.43°, which prolongs the distance between the contact corner and the metal edge (Figure 4b), where the AlGaN barrier is not recessed, and higher 2DEG density remains, while for the UTB ones, the different contacting shape effectively reduces the contacting distance and thus a higher tunneling probability. It is such a difference that makes the reduced $L_T$ as well as $\varrho_c$ in the UTB-AlGaN/GaN heterostructure with the 'ohmic-before-passivation' process.

EDS mapping and line scan results shown in Figure 4 clearly reveal Ti, Al, and Au diffusion to AlGaN, Ga, and N diffusion into Ti. It has been reported that a high concentration of N vacancy in i-AlGaN is helpful for ohmic-contact formation. For UTB ohmic contact, there is more Al enrichment at the edge of the ohmic metal stack, potentially forming AlN, resulting in significantly higher $\Phi_B$ compared to CB ones. On the other hand, the AlN at the edge of the metal stack is helpful in enhancing the density of 2DEG and contributing to a higher $N_D$. Additionally, since Au did not spread to the interface between AlGaN and GaN without connection with 2DEG, the ohmic-contact mechanism is TFE.

To make it clear, the different physical mechanism of ohmic contacts between UTB- and CB-AlGaN/GaN heterostructure is illustrated in Figure 5. Due to the large recess angle, there is a non-fully depletion area below the edge of the ohmic-contact area in the CB-AlGaN/GaN heterostructure shown in Figure 5d, where the density of 2DEG is less than that outside the contact area, resulting in a higher contact resistivity in AlGaN/GaN HEMTs, while as shown in Figure 5c, the density of 2DEG at the edge of the ohmic-contact area is abrupt for the recess-free ohmic contact on the UTB-AlGaN/GaN heterostructure, contributing to the enhanced $N_D$ and lower $R_c$. The effect of barrier layer thickness on 2DEG concentration has been described in detail in our previous study [20].

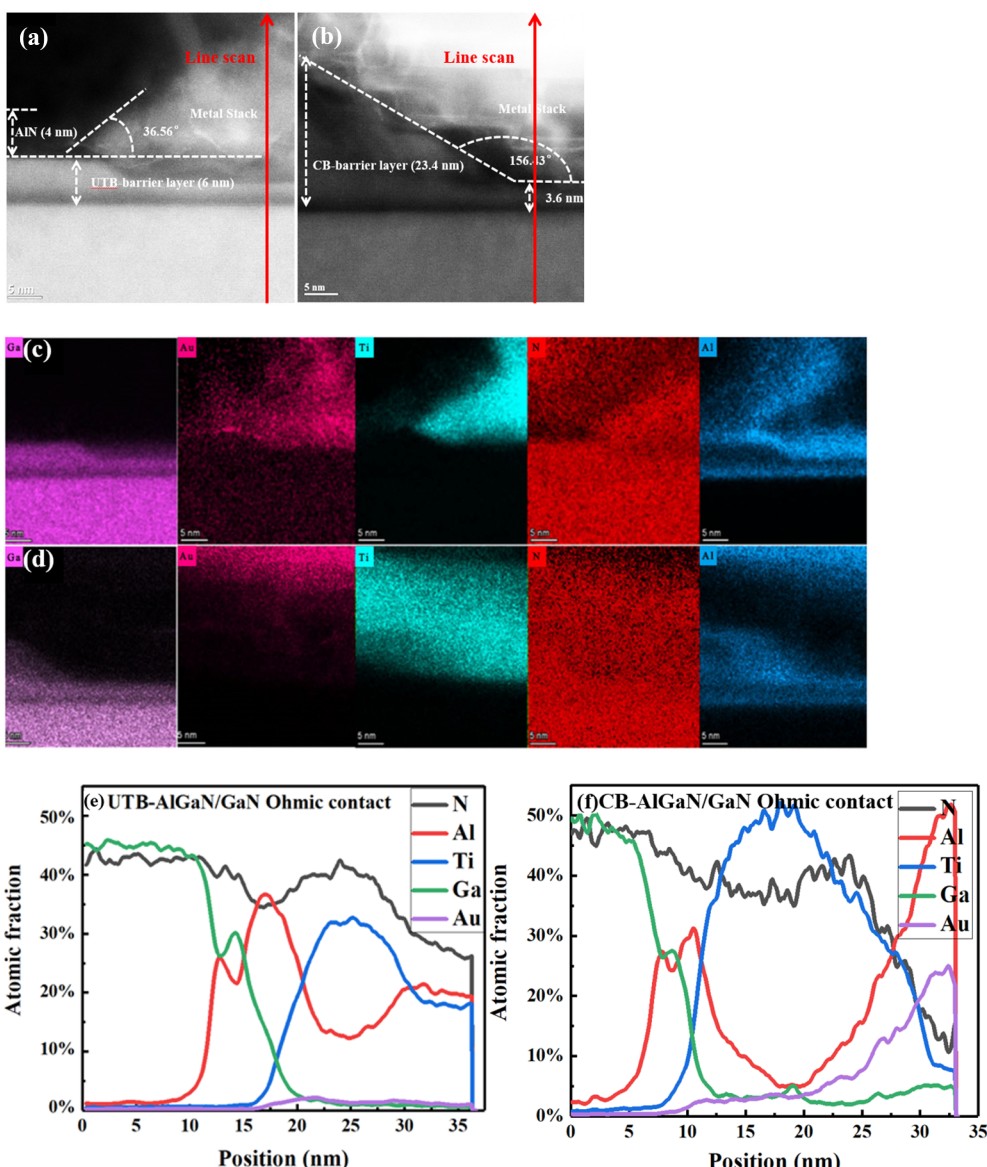

**Figure 4.** TEM cross−sectional view on (**a**) UTB and (**b**) CB−AlGaN/GaN heterostructure; EDS mapping of the ohmic contact on (**c**) UTB and (**d**) CB−AlGaN/GaN heterostructure; Line scan of the ohmic contact on (**e**) UTB and (**f**) CB−AlGaN/GaN heterostructure.

Figure 6a shows the output characteristics of the device with $L_{GS}$ = $L_{GD}$ = 1 μm based on UTB-AlGaN/GaN heterostructure with PEALD-AlN/PECVD-SiN$_x$ passivation. The maximum current density is about 750 mA/mm with $V_g$ = 3 V. The $R_{ON}$ extracted is also low, with a value of about 3.9 Ω·mm. The threshold voltage of the device is about −23 V due to the thick dielectric, which can be optimized by etching the dielectric below the gate. To prove that the well-optimized ohmic contact minimizes the device-specific $R_{ON}$, the gate transmission line method (GTLM) is adopted to extract the proportion of each part that makes up $R_{ON}$. Figure 6 shows the $L_G$ from 4 to 46 μm dependence of $R_{ON}$. The resistance of channel ($R_{ch}$) and access region ($R_{ac}$) is extracted to be 2.3 Ω·mm and 1.28 Ω·mm. It can be seen from Figure 7 that the optimized $R_c$ accounts for a small percentage of $R_{ON}$. Therefore, it can be shown that optimizing the ohmic-contact resistance has a great effect on the improvement of device performance.

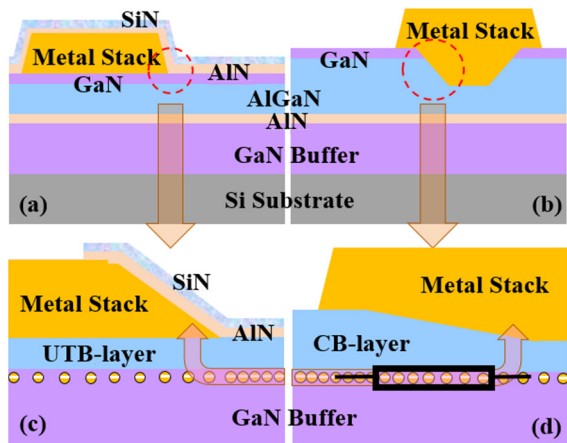

**Figure 5.** Schematic contact mechanisms of ohmic contacts to (**a**) UTB−AlGaN/GaN heterostructure and (**b**) CB−AlGaN/GaN heterostructure. Different distribution of 2DEG between (**c**) UTB−AlGaN/GaN heterostructure and (**d**) CB−AlGaN/GaN heterostructure.

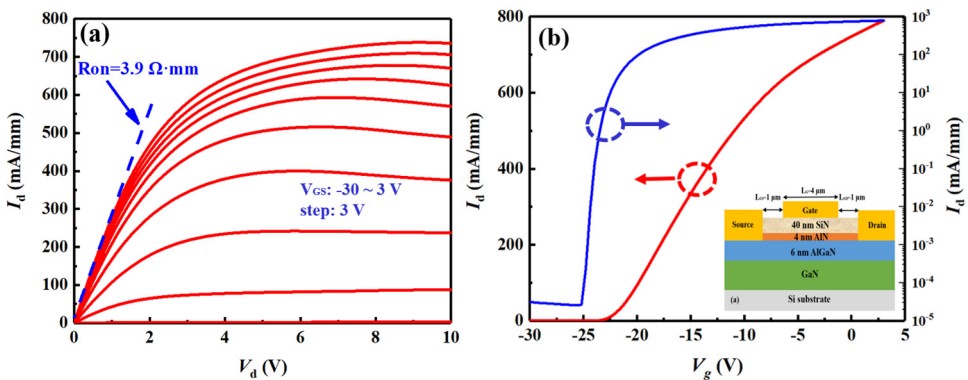

**Figure 6.** (**a**) Output characteristics and (**b**) transfer characteristics of the device based on UTB−AlGaN/GaN heterostructure with PEALD−AlN/PECVD−SiN$_x$ passivation.

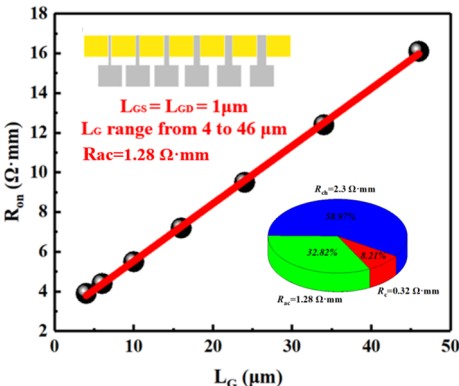

**Figure 7.** Evolution of R$_{ON}$ with L$_G$ and contribution of the resistance of each part to R$_{ON}$ of UTB−AlGaN/GaN heterostructure with PEALD−AlN/PECVD−SiN$_x$ passivation.

## 4. Conclusions

In summary, the ohmic-contact resistance to UTB-AlGaN/GaN heterostructure was effectively reduced via an 'ohmic-before-passivation' process, in which an alloyed Ti/Al/Ni/Au ohmic metal was formed prior to PEALD-AlN/PECVD-SiN$_x$ passivation used for recovery of 2DEG adjacent to the ohmic contact. The separation between the recovered 2DEG and the ohmic-contacting edge was remarkably reduced via the developed process, contributing to a reduced transfer length and low R$_c$ of 0.16 Ω·mm, as compared to that of ohmic contact

to the CB-AlGaN/GaN heterostructure with pre-ohmic recess. The low on-resistance of 3.9 $\Omega \cdot$mm and the the maximum current density of 750 mA/mm with $V_g$ = 3 V were achieved on the devices with the optimized ohmic contact. The non-recessed ohmic contact with the 'ohmic-before-passivation' process is a promising method to optimize the performance for the fabrication of low-voltage GaN-based power devices

**Author Contributions:** This research specifies the following individual contributions: Conceptualization, Y.J., J.F. and H.Y.; formal analysis, R.Z.; funding acquisition, S.H. and Q.J.; investigation, K.W.; methodology, Y.J.; project administration, X.W.; resources, X.L.; software, Y.Z.; supervision, S.H. and Q.J.; validation, Y.J.; visualization, S.H. and Q.J.; writing—original draft, Y.J. and S.H.; Writing—review and editing, Y.J. and S.H. All authors have read and agreed to the published version of the manuscript.

**Funding:** This work was supported in part by the National Key Research and Development Program of China under Grant 2022YFB3604400; in part by the Youth Innovation Promotion Association of Chinese Academy Sciences (CAS); in part by CAS-Croucher Funding Scheme under Grant CAS22801; in part by National Natural Science Foundation of China under Grant 62074161, Grant 62004213, and Grant U20A20208; in part by the Beijing Municipal Science and Technology Commission project under Grant Z201100008420009 and Grant Z211100007921018; in part by the University of CAS; and in part by IMECAS HKUST Joint Laboratory of Microelectronics.

**Data Availability Statement:** Not applicable.

**Conflicts of Interest:** The authors affirm that they do not have any conflicting priorities related to the research.

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
