# Peer review of "Optimization of Ohmic Contact to Ultrathin-Barrier AlGaN/GaN Heterostructure via an ‘Ohmic-Before-Passivation’ Process"

_electronics, doi:10.3390/electronics12081767_

Round 1
Reviewer 1 Report
Specific comments and recommendations:
1) The abstract of this work is too generally written. In my opinion, the authors have briefly stated the problem and then describe the work done. It is good to indicate more specific results achieved in this work;
2) The authors have to describe in more detail the field of application of the proposed method. Section 2 has to be presented in more detail, including illustrations of the layering method. Also, the authors have also to present some analytical formulas for the required structural and electrical parameters that would assist potential readers;
3) Is there a dependence of the resistance of the ohmic contact on the amplitude of the applied voltage, i.e. the so-called non-linear dependence of contact resistance? Is there a relationship between the contact parameters from the signal frequency?
Reviewer 2 Report
This manuscript describes the optimization of ohmic contact to UTB AlGaN/GaN heterostructure with the demonstration of how device is fabricated, measurement experiment is set up and material is characterized with TEM and EDS. Overall the results presented in this manuscript are interesting while there are some minor revisions needed before publication:
1. In section #2, authors described how device was fabricated with detailed process steps. Can authors explain a bit more on the annealing part (810C for 50s)? What is the purpose of the annealing given that there are different types of metal deposited as contact layer. Would it impact some of the metals like aluminum and gold which don't necessarily require such high annealing temp?
2. In section #3, authors mentioned that the improved ohmic contact can be explained by TFE with fitted Schottky barrier height and doping concentration. Can authors add the heterojucntion band diagrams for both structures? And please add the thermionic field emission formula to support the result modeling.
3. Generally Schottky barrier height is defined by the difference between the work function of the metal and electron affinity of semiconductor which should be fixed value. Can authors elaborate more on "The lower phi_B in the CB sample, as compared with that of the un-recessed UTB ones, may result from recess enhanced interface reaction between ohmic metal and AlGaN/GaN heterostructure"? What interface reaction impacted the the barrier height?
Reviewer 3 Report
I–V characteristics of ohmic contacts to various heterostructures are computed. The measurement data is fitted for various samples and the dependence of the resistivity on the temperature is identified. TEM cross-sectional view is provided for various scenarios.
The paper elaborates an interesting topic and deserves publication at MDPI Electronics, as long as the following points are discussed:
(A) Field distributions are necessary to see how the signal varies through the layers.
(B) What are the levels of absorption in such a resistive structure? [1,2]
(C) A weak point of the study is the total absence of analytical methods. The authors should discuss the possibility of treating the equipment into the layered structures with use of analytical methods [3,4].
[1] Intersubband absorption in GaN nanowire heterostructures at mid-infrared wavelengths, Nanotechnology, 2018.
[2] Wide-angle absorption of visible light from simple bilayers, Applied Optics, 2017.
[3] Integral equation analysis of a low-profile receiving planar microstrip antenna with a cloaking superstrate, Radio Science, 2012.
[4] Efficient Full-Wave Simulation in Layered, Lossy Media, Proceedings of the IEEE 1998 Custom Integrated Circuits Conference.
Round 2
Reviewer 1 Report
In this version of the manuscript, the authors have implemented the notes and recommendations. I propose that this manuscript be accepted for publication in a scientific journal.